# Fabrication of Multilayer Molds by Dry Film Photoresist

**DOI:** 10.3390/mi13101583

**Published:** 2022-09-23

**Authors:** Narek E. Koucherian, Shijun Yan, Elliot E. Hui

**Affiliations:** Department of Biomedical Engineering, University of California, Irvine, CA 92697, USA

**Keywords:** photoresist, soft lithography, microfluidics

## Abstract

Dry film photoresists are widely employed to fabricate high-aspect-ratio microstructures, such as molds for microfluidic devices. Unlike liquid resists, such as SU-8, dry films do not require a cleanroom facility, and it is straightforward to prepare uniform and reproducible films as thick as 500 µm. Multilayer patterning, however, can be problematic with dry film resists even though it is critical for a number of microfluidic devices. Layer-to-layer mask alignment typically requires the first layer to be fully developed, making the pattern visible, before applying and patterning the second layer. While a liquid resist can flow over the topography of previous layers, this is not the case with dry film lamination. We found that post-exposure baking of dry film photoresists can preserve a flat topography while revealing an image of the patterned features that is suitable for alignment to the next layer. We demonstrate the use of this technique with two different types of dry film resist to fabricate master molds for a hydrophoresis size-sorting device and a cell chemotaxis device.

## 1. Introduction

Microfluidic devices are commonly produced by preparing a master mold from which elastomeric devices can be repeatedly cast. Such molds are typically prepared by photolithography through the use of thick photoresists, such as the SU-8 family of epoxy resists [1]. These liquid resists are dispensed by spin coating, a process that is usually performed in a cleanroom to avoid particulate contamination. It can be challenging to prepare thick resist layers in the range of hundreds of micrometers with accurate and uniform thickness, free of cracks and defects. To address these limitations, thick photoresists are also commercially available as dry films that can be applied by lamination with low-cost equipment and no cleanroom. ADEX and SUEX are dry film photoresists based on the same chemistry as liquid SU-8 resists and are available at thicknesses ranging from 5 µm to over 500 µm [2,3]. While the cost per wafer is higher than with liquid resists, the capital costs to perform dry film lamination are far lower and the failure rate due to defects is substantially improved [4,5,6].

Microfluidic devices sometimes require a multilayer mold architecture to achieve their functionality. Two such examples are multi-chamber neural chemotaxis devices [7,8] and hydrophoretic size-based cell sorting devices [9,10], which both require two layers. Accurate layer-to-layer alignment is critical. This is usually accomplished by including alignment features on each mask layer and orienting each photomask to line up with markers on previous layers. With liquid resists, this is straightforward, as each layer can be fully developed before applying the next, rendering the features clearly visible during mask alignment. This is possible because liquid resists can flow over the topography of a patterned layer to replanarize the surface, and this remains true for the thick resist layers employed in microfluidics [11]. Lamination onto patterned dry film resist is more challenging, however, risking damage to the underlying features and poor layer-to-layer adhesion. Lamination onto patterned layers may be possible if only a small percentage of the underlying resist layer is removed during development, such as when microfluidic devices are directly constructed out of dry film resist [12,13,14,15,16,17,18,19]. However, microfluidic molds typically require most of each resist layer to be removed during development. Multilayer lamination thus becomes problematic, and fabrication of multilayer microfluidic molds by dry film resist has not previously been reported.

In this work, we found that post-exposure baking of ADEX or SUEX films following mask exposure produces a visible image suitable for mask alignment of subsequent layers, without the use of any developer. Thus, each resist layer can be laminated onto fully intact underlying layers. Once all layers have been laminated and exposed, developer can be applied to remove all unexposed regions at once. Alignment accuracy of about 50 µm was achieved by performing mask alignment by hand, using alignment markers of different sizes to facilitate both rough alignment by eye as well as fine alignment under a microscope. We demonstrate the use of this technique to fabricate master molds for a neural chemotaxis device, using the thinner ADEX dry film resist, as well as a scaled up hydrophoresis device, using the thicker SUEX dry film resist.

## 2. Materials and Methods

Photomasks were designed on AutoCAD (Autodesk, San Francisco, CA, USA) and printed on mylar sheets with a minimum feature resolution of 8 µm [20].

Lamination and photopatterning of dry film photoresists were performed as previously described [5], with additional steps as detailed below to achieve aligned multilayer patterning. Exposure was performed with a 365 nm LED source (M365LP1, ThorLabs, Newton, NJ, USA) placed 77 mm above the wafer. The UV intensity was measured to be 15.7 mW/cm^2^ at the wafer center. Lamination, mask exposure, and post-exposure bake were sequentially performed for each layer without using developer. First layer features were visible following post-exposure bake, facilitating alignment of the second layer mask. A layer of glass was placed on top of the printed photomask and affixed with binder clips to maintain alignment during the exposure process.

Post-exposure bake for ADEX wafers was performed by inserting exposed wafers directly into an oven preheated to 95 °C. After the required duration (Table 1), the oven was switched off and allowed to cool to room temperature. However, rapid temperature changes during the post-exposure bake were observed to produce deformation of SUEX features (Figure 1). Consequently, exposed SUEX wafers were inserted into the oven at 50 °C and the temperature was increased to 85 °C over 30 min. After remaining at 85 °C for an hour, the oven was slowly cooled to room temperature at a rate of 20 °C per hour.

After completing mask exposure and post-exposure bake for the final layer, all layers were developed at once. ADEX resists were used for thinner layers (5–75 µm) and SUEX resists were used for thicker layers (20–500 µm), but ADEX and SUEX were never combined in a single device since ADEX is developed in cyclohexanone and SUEX is developed in propylene glycol methyl ether acetate (PGMEA). In the neural chemotaxis device, two 50 µm layers of ADEX were sequentially laminated and then exposed together to obtain a 100 µm second layer.

Wafers were developed by immersion in cyclohexanone or PGMEA for 30–60 s, followed by rinsing in isopropyl alcohol (IPA). Dissolved photoresist was visible as white streak marks during the IPA rinse, which was continued until the streaks disappeared. Rounds of developing and rinsing were repeated until the white streak marks no longer appeared. A range of 10–20 rinse cycles was typically required. This process served to avoid overdeveloping the resist, which can result in the delamination of features. Delamination may also result from overexposure of small features or excessive rinse pressure. After development, wafers were gently washed with deionized water and dried using a nitrogen gun.

Mold fabrication was completed with a hard bake, as detailed in Table 1. Device structures were less sensitive to deformation in this step compared to the post-exposure bake, and hard baking was sometimes able to fix partially delaminated structures.

Devices were fabricated by standard soft lithography. Briefly, polydimethylsiloxane (PDMS, Sylgard 184, Dow Corning, Midland, MI, USA) was mixed at a 10:1 ratio of base to curing agent and degassed. PDMS was then poured over the mold, degassed for 1 h, and baked at 65 °C for 3 h. Molded PDMS parts were removed, punched with a biopsy needle to form ports, and finally sealed to a glass slide by plasma bonding.

## 3. Results

### 3.1. Post-Exposure Bake

We observed that patterned features in the dry film photoresist were revealed by post-exposure baking without the use of a developer (Figure 1). Thus, the first resist layer was left fully intact to support lamination of the second resist layer, while the second mask layer could still be aligned to visible features on the first layer. However, SUEX resist was found to be very sensitive to rapid temperature changes during the post-exposure bake, resulting in warped features (Figure 1), and so the process was optimized through a series of trials as detailed below. The results of these three trials are presented in Figure 1, showing that the best results required gradual heating and cooling.

Trial 1 (3 min UV exposure):Preheat oven to 85 °C.Bake wafer for 30 min at 85 °C.Turn off oven, allowing wafer to cool to 40 °C over 2–3 h.Trial 2 (3.5 min UV exposure):Preheat oven to 95 °C.Bake wafer for 30 min at 95 °C.Turn off oven, wait 2–3 h for oven to reach 40 °C, then remove wafer.Trial 3 (4 min UV exposure):Preheat oven to 50 °C.Bake wafer for 5 min at 50 °C.Set oven to 65 °C and bake wafer for 10 min.Set oven to 80 °C and bake wafer for 10 min.Set oven to 85 °C and bake wafer for 1 h.Set oven to 65 °C and bake wafer for 1 h.Set oven to 45 °C and bake wafer for 1 h.Turn off oven, wait 1 h for oven to reach 30 °C, then remove wafer.

### 3.2. Mask Alignment

Following mask exposure and post-exposure bake of the first resist layer, patterned features were rendered sufficiently visible to perform mask alignment to the second mask layer. This remained true after lamination of the second resist layer (Figure 2a), although the thinner resist on the chemotaxis device (100 µm ADEX) was clearer than the thicker resist on the hydrophoresis device (300 µm SUEX).

Following lamination of the second resist layer, the second layer photomask was placed over the wafer and flattened with a glass plate. The mask was first aligned roughly by eye (Figure 2b) and then fine alignment was performed under a microscope (Figure 2c). The aligned mask was held in place by binder clips, and UV exposure was performed. Following post-exposure bake and development, the final multilayer features were revealed. A layer-to-layer alignment accuracy of less than 50 µm was typically achieved (Figure 2d). Simultaneous horizontal, vertical, and angular alignment was ensured by employing alignment marks at the four corners of the wafer (Appendix A).

### 3.3. Device Specifications

Two multilayer devices were demonstrated. The first is a chemotaxis device designed for neuronal coculture [7,8]. The device consists of two chambers separated by narrow channels designed to study the migration of microglia from the outer chamber towards neurons and astrocytes in the inner chamber. A shallow height of 10 µm is required for the channels, while the two chambers should be larger to hold adequate culture media. We selected a 10 µm layer of ADEX resist for the channels and two 50 µm layers of ADEX for the chambers. It is important to note that UV exposure of second layer features also crosslinks the underlying resist on the first layer. Thus, this choice of layer heights produces 110 µm cell culture chambers. The inner chamber diameter was 3.8 mm, while the outer chamber diameter was 8 mm (Figure 3a). The channels connecting the two chambers were chosen to be 50 µm wide and 100 µm long but were drawn much longer on the mask pattern (blue, Figure 3a) to provide robustness against misalignment. Exposed areas on the first layer can be exposed a second time in overlapping mask regions but the resist is fully crosslinked regardless of whether it exposed once or twice. 

The second device is a hydrophoresis device designed for size-based sorting [9,10]. In hydrophoresis, slanted ridges patterned inside a microfluidic channel produce angled rotational flows that induce a size-dependent lateral displacement on particles traveling along the channel. Previous examples of hydrophoresis were designed for applications such as focusing red blood cells [9] or sorting platelets from whole blood [10]. Here, we sought to scale up the system to sort larger objects, with a total channel height of 550 µm and a ridge height of 300 µm, allowing particles up to 250 µm diameter to pass. We thus selected a first layer of 250 µm SUEX resist and second layer of 300 µm SUEX resist. Ridges were angled at 30° in the first part of the channel and then at 10° for the remainder of the channel (Figure 3b). The channel was 800 µm wide and 56 mm long. 

### 3.4. Multilayer Molds and Devices

After photopatterning of the second resist layer and post-exposure bake, the wafers were ready to be developed. All patterned layers were developed at the same time, after every layer had been laminated and exposed. Following a final hard bake, fabrication of the master molds was complete. The molds were employed to cast PDMS replicas, which were then punched to form ports and plasma bonded to glass. Completed neuronal chemotaxis molds and devices can be seen in Figure 4 and completed hydrophoresis molds and devices are shown in Figure 5.

## 4. Discussion and Conclusions

Dry film photoresists have numerous advantages for the fabrication of microfluidic master molds, particularly when cleanroom facilities are not available or when very thick resist layers are required. During parts of the COVID-19 pandemic, our research group was not able to access our campus cleanroom facility, and we came to rely heavily on dry film photolithography performed in our own laboratory. Dry film lithography has also served a critical role in allowing a wheelchair user in our research group to perform microfabrication despite challenges related to cleanroom access. 

Up to this point, however, it has not been possible to make multilayer microfluidic molds with dry film resist, even though a significant number of devices require the additional complexity offered by this approach. This work presents a straightforward method to accomplish multilayer patterning of dry film photoresists that should be simple to adopt. We demonstrate here that the process is compatible with either the ADEX or SUEX lines of dry film photoresist, enabling a wide variety of devices to be produced.

## Figures and Tables

**Figure 1 micromachines-13-01583-f001:**
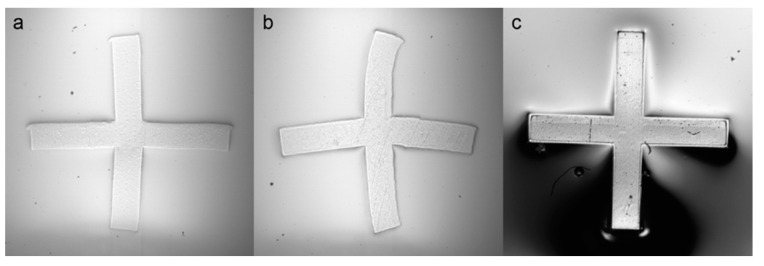
Results of post-exposure bake trials to study SUEX distortion: (**a**) Trial 1, (**b**) Trial 2, and (**c**) Trial 3. Distortion is minimized in Trial 3 by slow changes in temperature.

**Figure 2 micromachines-13-01583-f002:**
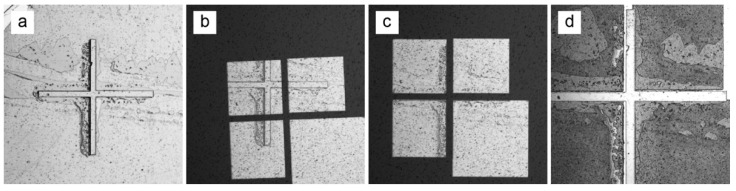
Mask alignment. (**a**) First layer features are visible through the second layer dry film photoresist. (**b**) Rough alignment of the second layer photomask by eye. (**c**) Fine photomask alignment performed under the microscope. (**d**) Following exposure and development, the final multilayer features are revealed. Here, the arms of the crosshairs are 125 µm wide, and thus misalignment is roughly 40 µm both horizontally and vertically. Scratches and smears are visible since the alignment marks were placed along the wafer edge and washing during development was focused on the center of the wafer where the functional devices are located.

**Figure 3 micromachines-13-01583-f003:**
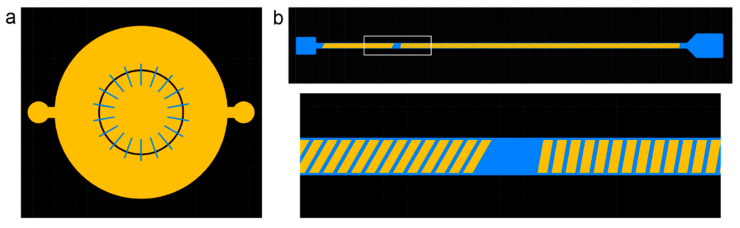
Dark-field mask layouts for chemotaxis and hydrophoresis devices. (**a**) Neuronal chemotaxis device consists of a 10 µm channel layer (blue) and a 100 µm chamber layer (orange). Note that the second layer mask pattern (orange) also crosslinks the first resist layer. (**b**) Hydrophoresis sorting device consists of a 250 µm channel layer (blue) and a 300 µm ridge layer (orange).

**Figure 4 micromachines-13-01583-f004:**
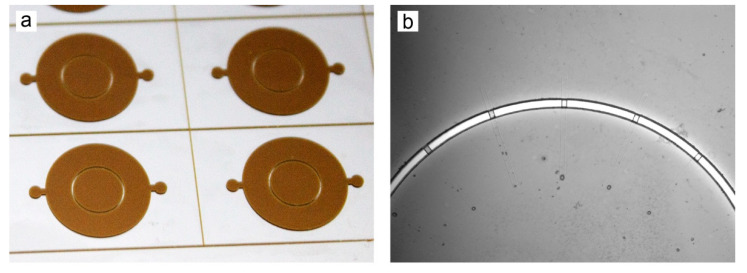
Neuronal chemotaxis device. (**a**) Multilayer master mold, following the final hard bake. (**b**) Microscope image of master mold. Cell migration channels traverse a wall separating the culture chambers. (**c**) Completed devices after molding PDMS, forming ports with a biopsy punch, and plasma bonding to a glass slide. (**d**) Microscope image of PDMS device, showing cell migration channels connecting the inner and outer cell culture chambers.

**Figure 5 micromachines-13-01583-f005:**
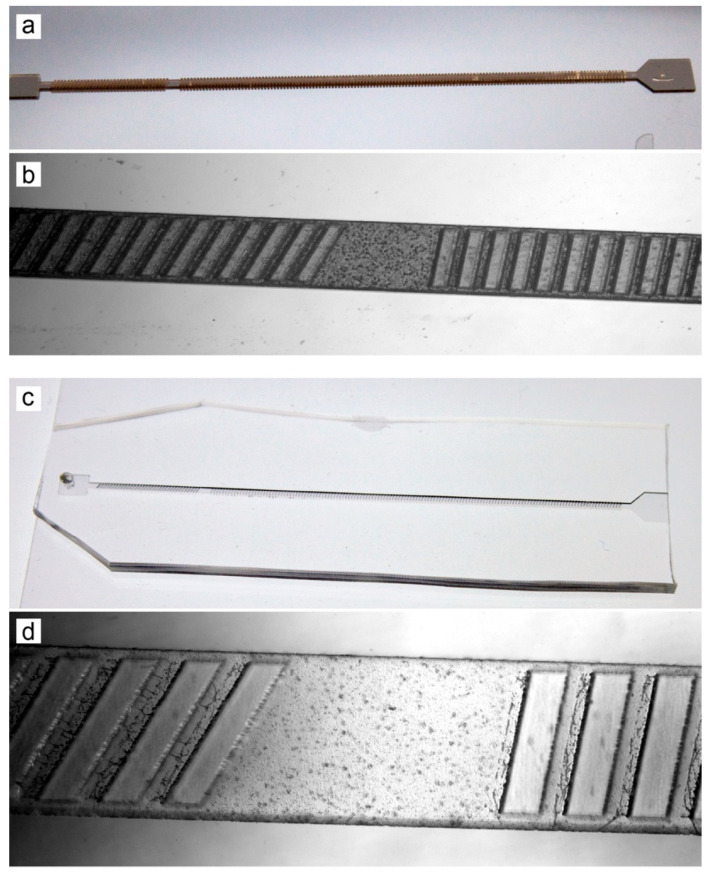
Hydrophoresis sorting device. (**a**) Multilayer master mold, following the final hard bake. (**b**) Microscope image of the master mold, showing excellent alignment of ridge features to the microchannel. (**c**) Completed hydrophoresis device after PDMS molding and plasma bonding to a glass slide. (**d**) Microscope image of a PDMS device, showing acceptable pattern fidelity to the mold.

**Table 1 micromachines-13-01583-t001:** Process parameters for various dry film photoresist types and layer thicknesses.

	Photoresist	Exposure	Post-Exposure Bake	Hard Bake
Chemotaxis Layer 1	10 µm ADEX	1 min	95 °C, 5 min	None
Chemotaxis Layer 2	2 × 50 µm ADEX	4 min	95 °C, 12 min	150 °C, 90 min
Hydrophoresis Layer 1	250 µm SUEX	4 min	85 °C, 60 min	None
Hydrophoresis Layer 2	300 µm SUEX	4.5 min	85 °C, 60 min	125 °C, 60 min

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
