# Peer review of "Fabrication of Multilayer Molds by Dry Film Photoresist"

_micromachines, 2022, doi:10.3390/mi13101583_

Round 1

Reviewer 1 Report

Fabrication of Multilayer Molds by Dry Film Photoresist

 Brief Summary: The authors highlight alignment as a significant challenge in producing multi-layer molds for casting of silicones using dry film photoresists. They demonstrate that partial baking of the first layer offers sufficient contrast for the subsequent layer to be aligned and exposed. This allows for casting of two multi-layer PDMS microfluidic devices.

General Comments:

The manuscript is well-written and the concept is well-communicated. In general the manuscript is under-cited and would strongly advise the authors to revisit and cite relevant literature (a cursory review of literature on multi-layer fabrication with DFR reveals notable uncited works: e.g. DOI: 10.1177/2472630319870126, https://doi.org/10.3390/mi12060632, a more in-depth search will most certainly find more relevant literature). A conclusions section should also be included.

Specific Comments:

1.       In line 13 of the abstract the authors describe the process as a ‘partial’ baking of the DFR, but list post exposure bake times for ADEX greater than those recommended by the ADEX data sheet. Saying ‘partial’ bake leads readers to believe the conditions are less than what would usually be used. (I couldn’t find recommended processing conditions for SUEX but I think the authors should clarify either way).

2.       The authors list only the exposure time and not intensity of the exposures in Table 1. I would recommend the addition of the exposure intensity.

3.       Could the authors add the number of iterations of development to line 90.

4.       Switch the order of the subsections in results section 3 to improve flow of the paper. Specifically move section 3.1 to after 3.3.

5.       Figure 1 is difficult to understand. Specifically does the yellow correspond to the 1st layer in the fabrication process (and bottom layer in the mold). I think a side view or text describing the vertical position of the channels would improve this figure. (If the blue layer is the first layer how does the structure in the left pane not float away after development – because it is only attached by a few small lines ?).

6.       The text from lines 137 to 159 would be better as a plot incorporated in Figure 2.

7.        I enjoy the personal anecdotes from lines 207 to 211 but I believe they are too informal for what is ultimately a conclusion section to this paper. I would recommend rewording this paragraph to acknowledge the improvements in accessibility afforded by DFRs and including personal anecdotes in a supplementary section (I would even like some pictures because you couldn’t swing a cat in our cleanroom let alone get a wheelchair in there).

8.       In lines 216 to 218 the authors say the process is compatible with both ADEX and SUEX, I think it is important to highlight it is not compatible with both at the same time.

Author Response

The manuscript is well-written and the concept is well-communicated. In general the manuscript is under-cited and would strongly advise the authors to revisit and cite relevant literature

The authors are grateful for the positive comments and detailed suggestions from Reviewer 1. After further literature review, we have added references to Wangler et al (2011), Johnson et al (2012), Mulloni et al (2019), and Cao et al (2021). Responses to specific reviewer comments are presented below:

1. In line 13 of the abstract the authors describe the process as a ‘partial’ baking of the DFR, but list post exposure bake times for ADEX greater than those recommended by the ADEX data sheet. Saying ‘partial’ bake leads readers to believe the conditions are less than what would usually be used.  

We have edited the abstract to replace “partial baking” with “post-exposure baking”. The partial bake terminology was meant to be in contrast to a hard bake, but we agree that it is confusing.

2. The authors list only the exposure time and not intensity of the exposures in Table 1.

The following text has been added: “Exposure was performed with a 365-nm LED source (M365LP1, ThorLabs, Newton, NJ, USA) placed 77 mm above the wafer. The UV intensity was measured to be 15.7 mW/cm2 at the wafer center.”

3. Could the authors add the number of iterations of development to line 90.

The following text has been added: “10-20 rinse cycles were typically required.”

4. Switch the order of the subsections in results section 3 to improve flow of the paper. Specifically move section 3.1 to after 3.3. 

Changing the order of the subsections is a good suggestion, and we have made the changes as recommended. “Track changes” was turned off during these edits in order not to obscure more specific edits in the text.

5. Figure 1 is difficult to understand.

The text has been modified as follows. (Fig. 1 is now Fig. 3, after reorganizing the sections as suggested.): “We selected a 10-µm layer of ADEX resist for the channels and two 50-µm layers of ADEX for the chambers. It is important to note that UV exposure of second-layer features will also crosslink underlying resist on the first layer. Thus, this choice of layer heights will produce 110-µm cell culture chambers. The inner chamber diameter was 3.8 mm, while the outer chamber diameter was 8 mm (Fig. 3a). The channels connecting the two chambers were chosen to be 50 µm wide and 100 µm long, but were drawn much longer on the mask pattern (blue, Fig. 3a) to provide robustness against misalignment. Exposed areas on the first layer will be exposed a second time in overlapping mask regions, but the resist is fully crosslinked regardless of whether it exposed once or twice.”

Additionally, the following text has been added to the Fig. 3 caption: “Note that the second-layer mask pattern (orange) also crosslinks the first resist layer.”

6. The text from lines 137 to 159 would be better as a plot incorporated in Figure 2.

Graphical presentation of the heating and cooling protocols in Trials 1-3 is not a bad idea and would offer a visual comparison between the three protocols. However, individual time blocks are as short as 10 minutes, over a total time course nearing 5 hours. We believe the textual description presents the details of the protocols most accurately.

7. I enjoy the personal anecdotes from lines 207 to 211 but I believe they are too informal for what is ultimately a conclusion section to this paper. I would recommend rewording this paragraph to acknowledge the improvements in accessibility afforded by DFRs and including personal anecdotes in a supplementary section (I would even like some pictures because you couldn’t swing a cat in our cleanroom let alone get a wheelchair in there).

We set up dry-film lamination and UV exposure in a fairly large microscope room after clearing out some other equipment. The seat of the powered wheelchair had the capability to elevate, allowing access to a chemical fume hood for wafer development. This student has graduated and the space has been reconfigured. Thank for your comment regarding the informal nature of the anecdotes. We considered your suggestions, but we believe that the current two-sentence format is the most efficient way to communicate the information. 

8. In lines 216 to 218 the authors say the process is compatible with both ADEX and SUEX, I think it is important to highlight it is not compatible with both at the same time.

The text has been amended as follows: “We have demonstrated that the process is compatible with either the ADEX or SUEX lines of dry film photoresist, enabling a wide variety of devices to be produced.”

Reviewer 2 Report

In this manuscript, authors introduced a fabrication method for multiplayer molds with high aspect ratio by dry film photoresist

Overall, this is a solid manuscript that proposes a new way of master mold fabrication without the need of conventional photolithography equipment setup. The capability described in the manuscript is important and will be valuable to the field of droplet microfluidics. The paper is well written, figures are quite good. The manuscript may be published as is.

Author Response

The authors greatly appreciate the positive comments from Reviewer 2.

Round 2

Reviewer 1 Report

Thank you for making these changes.